# Routine measurement of cardiometabolic disease risk factors in primary care in England before, during, and after the COVID-19 pandemic: A population-based cohort study

Frederick K. Ho[1‡*], Caroline Dale[2‡], Mehrdad A. Mizani[3], Thomas Bolton[3], Ewan R. Pearson[3,4], Jonathan Valabhji[5], Christian Delles[6], Paul Welsh[6], Shinya Nakada[1], Daniel Mackay[1], Jill P. Pell[1], Chris Tomlinson[7], Steffen E. Petersen[3,8], Benjamin Bray[9], Mark Ashworth[10], Kazem Rahimi[11], Mamas Mamas[12], Julian Halcox[13], Cathie Sudlow[4], Reecha Sofat[2,4‡], Naveed Sattar[7‡*], CVD-COVID-UK/COVID-IMPACT Consortium[¶]

1 School of Health and Wellbeing, University of Glasgow, Glasgow, United Kingdom, 2 Department of Pharmacology & Therapeutics, Institute of Systems, Molecular and Integrative Biology, University of Liverpool, Liverpool, United Kingdom, 3 British Heart Foundation Data Science Centre, Health Data Research UK, London, United Kingdom, 4 Division of Population Health & Genomics, University of Dundee, Dundee, United Kingdom, 5 Division of Metabolism, Digestion and Reproduction, Imperial College London, London, United Kingdom, 6 School of Cardiovascular and Metabolic Health, University of Glasgow, Glasgow, United Kingdom, 7 Institute of Health Informatics, University College London, London, United Kingdom, 8 The William Harvey Research Institute, Queen Mary University of London, London, United Kingdom, 9 School of Population Health and Environmental Sciences, King's College London, London, United Kingdom, 10 Department of Population Health Sciences, King's College London, London, United Kingdom, 11 Nuffield Department of Women's & Reproductive Health, University of Oxford, Oxford, United Kingdom, 12 School of Medicine, Keele University, Keele, United Kingdom, 13 Swansea University Medical School, Faculty of Medicine, Health and Life Science, Swansea University, Swansea, United Kingdom

‡ FKH and CD share first authorship on this work. RS and NS are joint senior authors on this work.
¶ Membership of CVD-COVID-UK/COVID-IMPACT Consortium is provided in the Acknowledgements.
* Frederick.Ho@glasgow.ac.uk (FKH); Naveed.Sattar@glasgow.ac.uk (NS)

**Data Availability Statement:** The data used in this study are available in NHS England's Secure Data Environment (SDE) service for England, but as restrictions apply they are not publicly available (https://digital.nhs.uk/services/secure-data-environment-service). The CVD-COVID-UK/COVID-

## Abstract

### Background

This study estimated to what extent the number of measurements of cardiometabolic risk factors (e.g., blood pressure, cholesterol, glycated haemoglobin) were impacted by the COVID-19 pandemic and whether these have recovered to expected levels.

### Methods and findings

A cohort of individuals aged ≥18 years in England with records in the primary care—COVID-19 General Practice Extraction Service Data for Pandemic Planning and Research (GDPPR) were identified. Their records of 12 risk factor measurements were extracted between November 2018 and March 2024. Number of measurements per 1,000 individuals were calculated by age group, sex, ethnicity, and area deprivation quintile. The observed number of measurements were compared to a composite expectation band, derived as the union of the 95% confidence intervals of 2 estimates: (1) a projected trend based on data

IMPACT programme led by the BHF Data Science Centre (https://bhfdatasciencecentre.org/) received approval to access data in NHS England's SDE service for England from the Independent Group Advising on the Release of Data (IGARD) (https://digital.nhs.uk/about-nhs-digital/corporate-information-and-documents/independent-group-advising-on-the-release-of-data) via an application made in the Data Access Request Service (DARS) Online system (ref. DARS-NIC-381078-Y9C5K) (https://digital.nhs.uk/services/data-access-request-service-dars/dars-products-and-services). The CVD-COVID-UK/COVID-IMPACT Approvals & Oversight Board (https://bhfdatasciencecentre.org/areas/cvd-COVID-uk-COVID-impact/) subsequently granted approval to this project to access the data within NHS England's SDE service for England. The de-identified data used in this study were made available to accredited researchers only as per the data sharing agreement and the ethical approval. Those wishing to gain access to the data should contact bhfdsc@hdruk.ac.uk in the first instance. A pre-specified analysis plan published on GitHub, along with the phenotyping and analysis code (https://github.com/BHFDSC/CCU008_01).

**Funding:** The British Heart Foundation Data Science Centre (grant no. SP/19/3/34678, awarded to Health Data Research (HDR) UK) funded co-development (with NHS England) of the Secure Data Environment service for England, provision of linked datasets, data access, user software licences, computational usage, and data management and wrangling support, with additional contributions from the HDR UK Data and Connectivity component of the UK Government Chief Scientific Adviser's National Core Studies programme to coordinate national COVID-19 priority research. Consortium partner organisations funded the time of contributing data analysts, biostatisticians, epidemiologists, and clinicians. The funders had no role in study design, data collection and analysis, decision to publish, or preparation of the manuscript.

**Competing interests:** NS has consulted for and/or received speaker honoraria from Abbott Laboratories, Amgen, AstraZeneca, Boehringer Ingelheim, Eli Lilly, Hanmi Pharmaceuticals, Janssen, Merck Sharp & Dohme, Novartis, Novo Nordisk, Pfizer, Roche Diagnostics, and Sanofi; and received grant support paid to his University from AstraZeneca, Boehringer Ingelheim, Novartis, and Roche Diagnostics outside the submitted work. JV was National Clinical Director for Diabetes and Obesity at NHS England from 2013 to September 2023. JH received research grants from Amgen, British Heart Foundation, Health and Care Research

prior to the COVID-19 pandemic; and (2) an assumed stable trend from before pandemic. Point estimates were calculated as the mid-point of the expectation band.

A cohort of 49,303,410 individuals aged ≥18 years were included. There was sharp drop in all measurements in March 2020 to February 2022, but overall recovered to the expected levels during March 2022 to February 2023 except for blood pressure, which had prolonged recovery. In March 2023 to March 2024, blood pressure measurements were below expectation by 16% (−19 per 1,000) overall, in people aged 18 to 39 (−23%; −18 per 1,000), 60 to 79 (−17%; −27 per 1,000), and ≥80 (−31%; −57 per 1,000). There was suggestion that recovery in blood pressure measurements was socioeconomically patterned. The second most deprived quintile had the highest deviation (−20%; −23 per 1,000) from expectation compared to least deprived quintile (−13%; −15 per 1,000).

## Conclusions

There was a substantial reduction in routine measurements of cardiometabolic risk factors following the COVID-19 pandemic, with variable recovery. The implications for missed diagnoses, worse prognosis, and health inequality are a concern.

## Author summary

### Why was this study done?

- Studies have shown that the initial COVID-19 restrictions were associated with a sharp drop in cardiometabolic risk factor measurements in primary care.

- However, the extent to which recovery has occurred until 2024 and how recovery varies by age, sex, ethnicity, or deprivation, remains unknown.

### What did the researchers do and find?

- We extracted data from the General Practice Extraction Service Data for Pandemic Planning and Research (GDPPR), which covers 98% general practices in England.

- Examining a cohort of over 49 million adults, we found that most of the risk factor measurements recovered to the expected level by 2022 to 2023.

- The recovery appeared to be socioeconomically patterned.

### What do these findings mean?

- The prolonged recovery of blood pressure measurement, consistent with findings from Health Survey for England 2021, could mean missed diagnoses and worse prognosis.

- The inequality in measurement recovery could also lead to exacerbated inequality to health outcomes.

Wales, and speaker honorarium from Amgen. CD is the Treasurer of European Council for Cardiovascular Research, Association of Physicians of Great Britain and Ireland, Council Member of the European Society of Hypertension, and the Vice President of Scottish Heart & Arterial disease Risk Prevention (SHARP). All other authors declared no potential conflicts of interests.

**Abbreviations:** ALT, alanine aminotransferase; AST, aspartate aminotransferase; BMI, body mass index; BP, blood pressure; CVD, cardiovascular disease; eGFR, estimated glomerular filtration rate; EHR, electronic health record; GAM, generalised additive model; GGT, gamma-glutamyltransferase; GPES, General Practice Extraction Service; HDL, high-density lipoprotein; IMD, index of multiple deprivation; LDL, low-density lipoprotein; LSOA, lower layer super output area; ONS, Office for National Statistics.

- It should be noted that a shorter period of retrospective data was used to establish the expected level of measurements, which might be less reliable, and that the deaths that occurred during COVID might have changed the population structure and therefore the need of risk factor measurements.

## Introduction

The COVID-19 pandemic had a significant impact on many aspects of health care. Prominent among these was cessation of routine face-to-face health checks, [1] designed to detect common chronic cardiometabolic conditions such as type 2 diabetes, hypertension, that increase the risk of cardiovascular disease (CVD). As a result, prescriptions for common preventative medicines fell substantially [2]. For example, there was a decline in the dispensing of antihypertensive medications between March 2020 and July 2021, with nearly half a million fewer individuals across England, Scotland, and Wales initiating treatment than expected. Modelling predicted that this decline could result in an excess of over 13,000 CVD events in Great Britain [2]. There is, therefore, an urgent need to identify and treat individuals with undetected CVD risk factors in order to avoid large numbers of excess future CVD events and progression to more severe forms of CVD. This requires optimal risk factor measurement across the population within the health service.

An OpenSafely study that utilised the NHS England data examined the trend of 11 indicators of general practice clinical activity before and after the COVID-19 pandemic, including cholesterol, glycated haemoglobin (HbA1c), and blood pressure (BP) monitoring [3]. They showed that there was a substantial drop in all these measurements up to December 2021. For example, there was a 2.3% drop in HbA1c, a 13% drop in cholesterol, and a 42% drop in BP measurement in 2021 compared to 2019. The authors classified the former two to be "recovered" to the pandemic level, while the latter had a "sustained drop." These largely coincided with other preprint and published papers [4–6]. However, none of these studies included a comprehensive sets of risk factors relevant to cardiometabolic disease, e.g., body mass index (BMI) and smoking, or examined whether any population subgroups were potentially disproportionately affected, important questions which could help direct future health initiatives.

Here, we used data on risk factor measurements in England to examine the pattern of cardiometabolic risk factor measurements over time, before, during, and after the pandemic up to early 2024. We aimed to determine whether risk factor ascertainment has recovered to expected levels and, if not, where gaps remain. Inequality in recovery by age, sex, ethnicity, and area deprivation were also examined. Given that models of care have changed, with less face-to-face clinical appointments, we hypothesised that some key measurements that require physical tests or blood draws—e.g., BMI, BP, and cholesterol—may still lag behind pre-pandemic levels. Such analyses are important to understand future disease patterns, potential missed opportunities for preventative care and healthcare provision, particularly in high-risk groups where virtual clinic appointments may not be optimal.

## Methods

### Databases and study population

This is a retrospective cohort study analysing primary care risk factor measurement data from the General Practice Extraction Service (GPES) extract Data for Pandemic Planning and Research (GDPPR), including data from 98% of all English general practices. GDPPR consists

of data from 4 primary care electronic health record (EHR) systems, including the 2 major ones: EMIS and TPP (99.5% records). The earliest data for selected risk factors in GDPPR were available from May 2018 and October 2018 in EMIS and TPP, respectively. We included risk factors measured from November 2018 to March 2024 to ensure data completeness.

This study included individuals with any records in the GDPPR, excluding those aged under 18 years on 1 November 2018, those with unknown recorded sex, those without a valid residential address in England, and those with less than 1 month of follow-up. Follow-up started 2 years before the first data reporting date, marking the time-based cut-off for selected risk factors according to the business rules of GDPPR. Follow-up ended at date of death (for those who died) or April 2024.

Individuals' death status was ascertained using the Office for National Statistics (ONS) Civil Registration of Death records, which is mandatory for all deaths in the UK. Individuals' latest residential address lower layer super output areas (LSOAs) to link with the area-based index of multiple deprivation (IMD version 2019). Their ethnicity was primarily ascertained using primary care data in GDPPR (83%) and supplemented from codes in secondary care data (11%) as 5 broad categories: White, Black, Asian, Mixed, and Other. The detailed methodology can be found in a published paper [7]. Individuals with unknown ethnicity (6%) were grouped with the "Other" group.

## Risk factor measurements

Risk factors included: BMI, smoking habit, alcohol consumption, BP, glycated haemoglobin (HbA1c), fasting glucose, total, low-density lipoprotein (LDL), and high-density lipoprotein (HDL) cholesterol, triglycerides, liver function tests (aspartate aminotransferase (AST), alanine aminotransferase (ALT), and gamma-glutamyltransferase (GGT)), and estimated glomerular filtration rate (eGFR). All factors apart from eGFR were prespecified based on their relevance to primary and secondary prevention to cardiometabolic disease. For example, BP, cholesterol, and LFTs [8] are all known causal risk factors for CVDs (or, in the case of LFTs, influence their management), HbA1c is a common marker for disease monitoring in diabetes, and kidney function decline is a common complication in people with obesity and diabetes [9]. eGFR was added to the analysis owing to emerging understanding of the importance of renal function in cardiometabolic health [10].

SNOMED-CT codes to ascertain these risk factors were based on the primary care business rules and reference set for GDPPR published by NHS England (https://digital.nhs.uk/data-and-information/data-collections-and-data-sets/data-collections/quality-and-outcomes-framework-qof/quality-and-outcome-framework-qof-business-rules/primary-care-domain-reference-set-portal). The codes defined by National Diabetes Audit were used for BMI, BP, smoking, and HbA1c. Relevant codes in the GDPPR reference set were utilised for other risk factors. We included all risk factors between November 2018 and March 2024. Any repeated measurements for the same code cluster on a single day for an individual were excluded to avoid duplicating the recording of that code. Except for smoking habits and alcohol consumption, all risk factor records without a value were excluded.

## Statistical analyses

All relevant risk factor measurements of the included individuals were extracted from the GDPPR. For each risk factor, the monthly number of measurements per 1,000 individuals was calculated by age group (18 to 39, 40 to 59, 60 to 79, ≥80 years), sex, ethnicity, and IMD quintile.

Both the absolute (number per 1,000 individuals) and relative (%) deviations from expected levels were calculated. Two strategies were used to estimate the expected level of number of measurements, based on generalised additive models (GAMs) [11] fitted on pre-pandemic data (November 2018 to February 2020). Quasi-Poisson distribution was used, with log-transformed eligible population size included as an offset variable. Age group, sex, ethnicity, and IMD quintile were modelled as categorical variables. Month trends were modelled using cyclical P-spline [12]. Linear long-term trend modelled based on an index variable denoting year difference from March 2020. For example, March 2020 is "0," March 2019 is "−1" (1 year prior), and May 2022 is "2.17" (2 + 2/12 years after). The interactions of the trend variable with age group, sex, ethnicity, and IMD quintile were included to allow trend differences by subgroup.

Because only 14 months of retrospective data is available, we constructed a composite expected level of measurements based on 2 projected trends. The first estimate used direct projection of the abovementioned models, assuming the trend estimated prior to be reliable and applicable. The second estimate assumed the long-term trend to be unchanged from March 2020, i.e., setting the trend variable as "0" in the model for projection. The projections from these 2 estimates were combined to create an expectation band as the union of the 95% confidence intervals estimated from the 2 methods. That is, the upper bound was the maximum of the 95% confidence upper bounds of both estimates, and the lower bound being minimum of the 95% confidence lower bounds. Point estimates were calculated as the mid-point of the expectation bands. This method allows a wider expectation band which provides a more conservative estimates in deviations. Absolute deviations were calculated as the difference between the observed and the expected values. Relative deviations were calculated as (observe–expected)/expected, expressed as %s. Three summary periods were used for reporting: March 2020 to February 2022 (during which COVID-19 restrictions were in place); March 2022 to February 2023; March 2023 to March 2024.

Data curation was completed in Databricks (11.3 LTS ML, Apache Spark 3.3.0) using PySpark, and analyses were conducted using R version 4.0.3 with the mgcv package.

The study was reported based on the STROBE checklist (S1 STROBE checklist). This analysis was performed according to a prespecified analysis plan published on GitHub, along with the phenotyping and analysis code (https://github.com/BHFDSC/CCU008_01). The protocol aimed to examine the absolute number of measurements. The current analysis examined number of measurements per 1,000 individuals to account for different eligibility period because of death.

### Patient and public involvement

Patients or the public were not involved in the design, or conduct, or reporting, or dissemination plans of our research.

### Ethical approvals

The North East—Newcastle and North Tyneside 2 research ethics committee provided ethical approval for the CVD-COVID-UK research program (REC no. 20/NE/0161) to access, within secure trusted research environments, unconsented, whole-population, de-identified data from EHRs collected as part of patients' routine healthcare.

### Results

A total of 66,007,910 individuals were identified in GDPPR, of which 16,447,895 aged <18 years on 1 November 2018 were excluded. A further 2,005, 248,230, and 6,370 were excluded

**Table 1. Characteristics of included individuals.**

|  | Number | % |
|---|---|---|
| **Total** | **49,303,410** | **100.0** |
| Age at 1 November 2018 |  |  |
|   18–39 | 20,520,300 | 41.6 |
|   40–59 | 15,980,950 | 32.4 |
|   60–79 | 10,345,990 | 21.0 |
|   80+ | 2,456,170 | 5.0 |
| Sex |  |  |
|   Female | 24,785,815 | 50.3 |
|   Male | 24,517,595 | 49.7 |
| Ethnicity |  |  |
|   White | 38,324,280 | 77.7 |
|   Asian | 5,423,070 | 11.0 |
|   Black | 2,082,980 | 4.2 |
|   Mixed | 829,610 | 1.7 |
|   Other/Unknown | 2,643,470 | 5.4 |
| IMD quintile |  |  |
|   1 –most deprived | 9,893,875 | 20.1 |
|   2 | 10,498,370 | 21.3 |
|   3 | 10,089,460 | 20.5 |
|   4 | 9,639,765 | 19.6 |
| 5 –least deprived | 9,181,940 | 18.6 |

Numbers are rounded to multiples of 5 to comply with NHS England's statistical disclosure control rules.

as their sex was not recorded, had no valid residential LSOA in England, and had less than 1 month of follow-up, respectively. This study included a total of 49,303,410 individuals (Fig A in S1 File). Characteristics of the included individuals are shown in Table 1. Over 40% of the included individuals aged 18 to 39 and only 5% aged ≥80. Female:male ratio was 50.3:49.7. Over three quarters (78%) of individuals were of White ethnicity, 11% were Asian, 4.2% were Black, 1.7% were mixed, and 5.4% were of other or unknown ethnicities. Approximately equal proportions were included from each IMD quintile.

The trends in measurements are shown in **Fig 1**. There was a sharp drop in all measurements in March 2020 but most recovered to the lower bounds of expected level by 2022, except for BP and fasting glucose.

The estimated % deviations of all measurements are shown in **Fig 2**, based on the GAMs shown in **Table A in S2 File**. The absolute and relative deviations over the three time periods, overall and by subgroups, are shown in Tables B and C in S2 File, respectively.

Deviations from expected levels varied by age subgroups and risk factors. For example, in March 2023 to March 2024, BP measurements were still below expectation by 16% (−19 per 1,000), in people aged 18 to 39 (−23%; −18 per 1,000), 60 to 79 (−17%; −27 per 1,000), and ≥80 (−31%; −57 per 1,000), but that was within expected band for people aged 40 to 59 (Tables B and C in S2 File). However, a different pattern is found in alcohol consumption, where younger people (aged 18 to 39) had increasing measurement, and HbA1c measurements far above expectation (Tables B and C in S2 File). Absolute and relative deviations over time are shown in Figs 3 and B–D in S1 File.

Females generally had more measurements than expected compared with males (Fig C in S1 File), except for BP measurements where female had a larger deficit (−19%; −28 per 1,000)

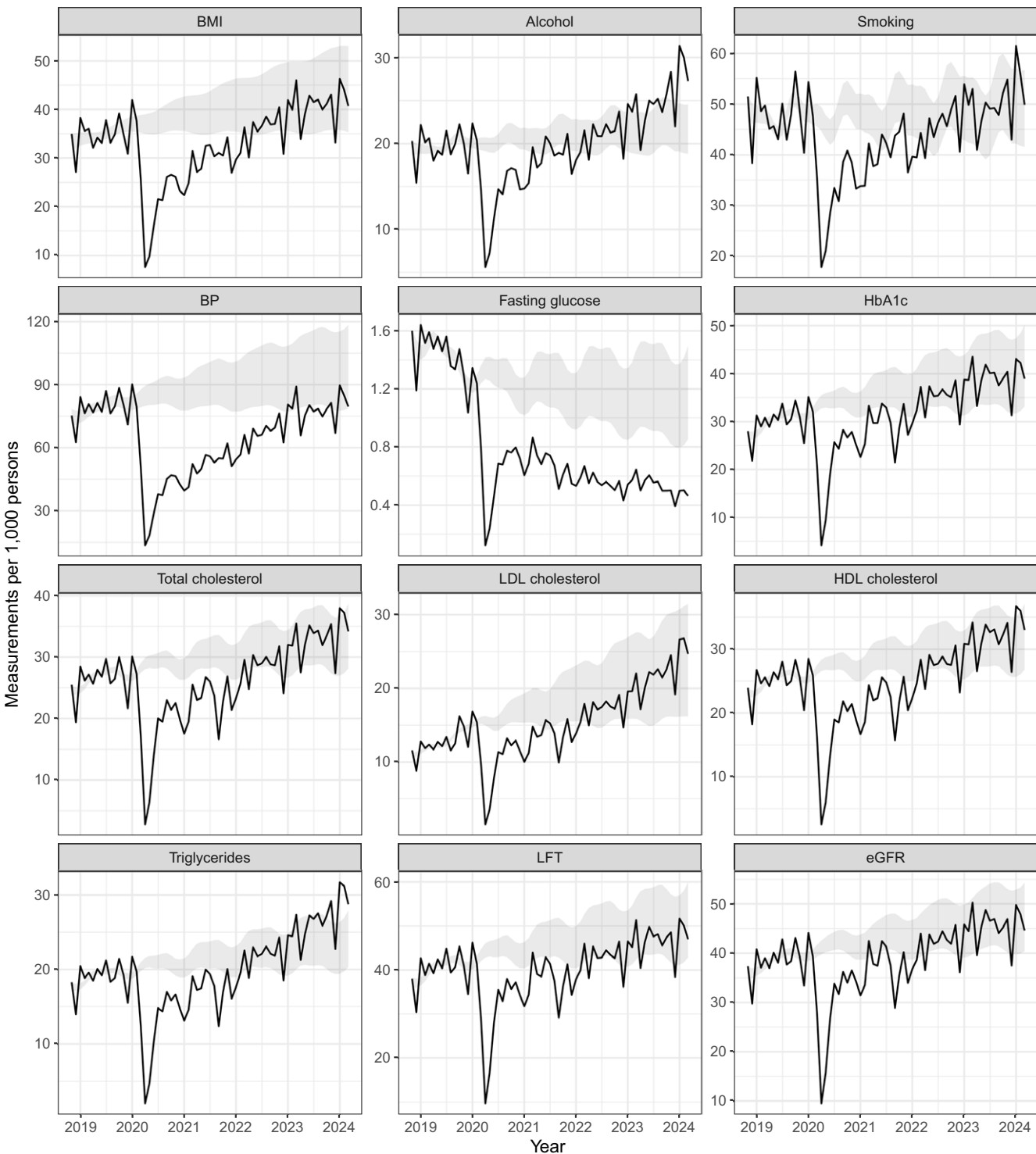

**Fig 1. Monthly number of measurements per 1,000 individuals between November 2018 and March 2024.** Shaded areas are ranges of expected levels, aggregated from the 95% CIs of projected trends based on data between November 2018 and February 2020, and assumed stable trends from February 2020. BMI, body mass index; BP, blood pressure; HbA1c, glycated haemoglobin; LDL, low-density lipoprotein; HDL, high-density lipoprotein; LFT, liver function test; eGFR: estimated glomerular filtration rate.

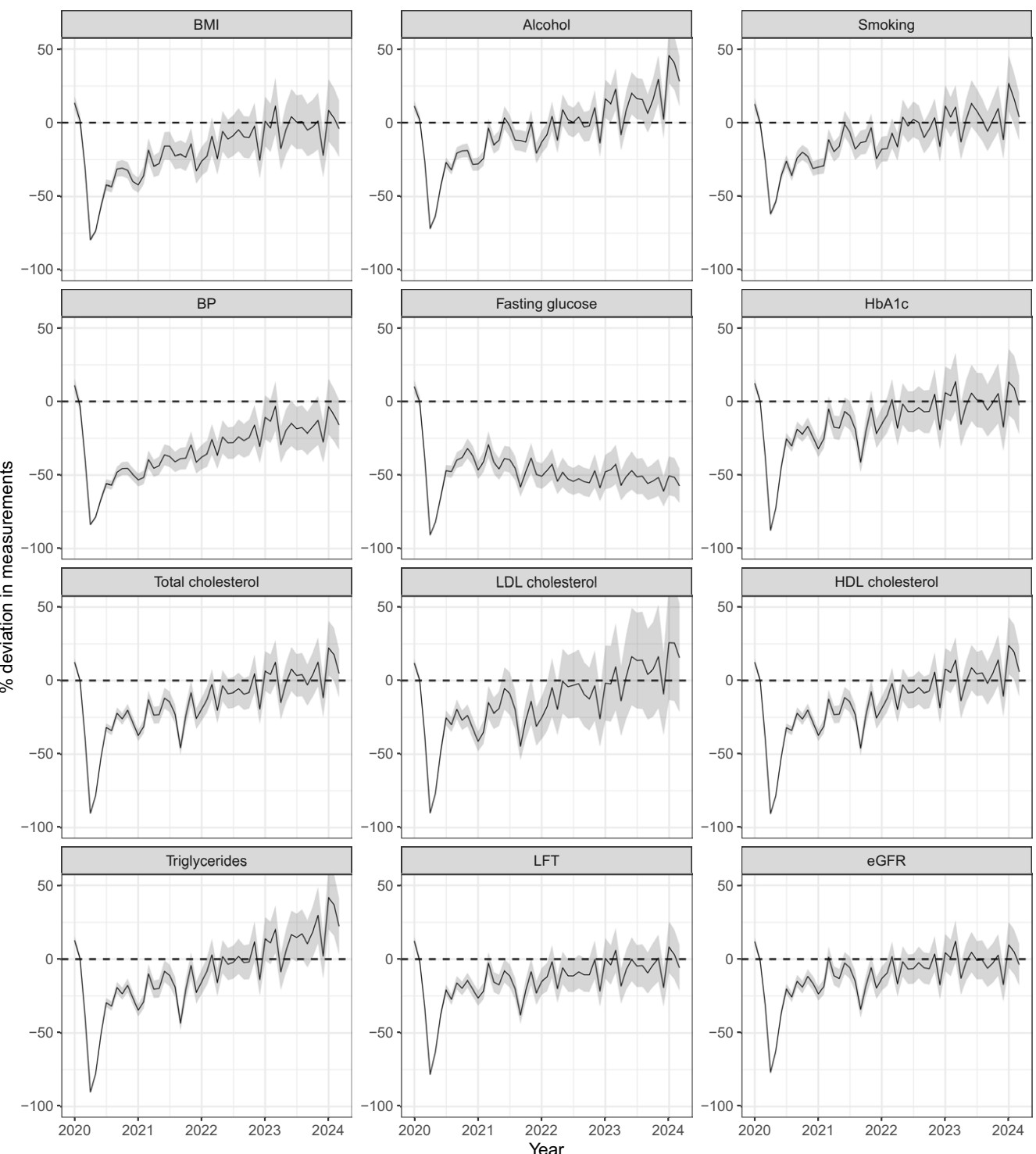

**Fig 2. Estimated % deviations of measurements between November 2018 and March 2024.** Horizontal dashed line indicates on difference from expected level. Shaded areas are composite of 95% confidence intervals. BMI, body mass index; BP, blood pressure; HbA1c, glycated haemoglobin; LDL, low-density lipoprotein; HDL, high-density lipoprotein; LFT, liver function test; eGFR, estimated glomerular filtration rate.

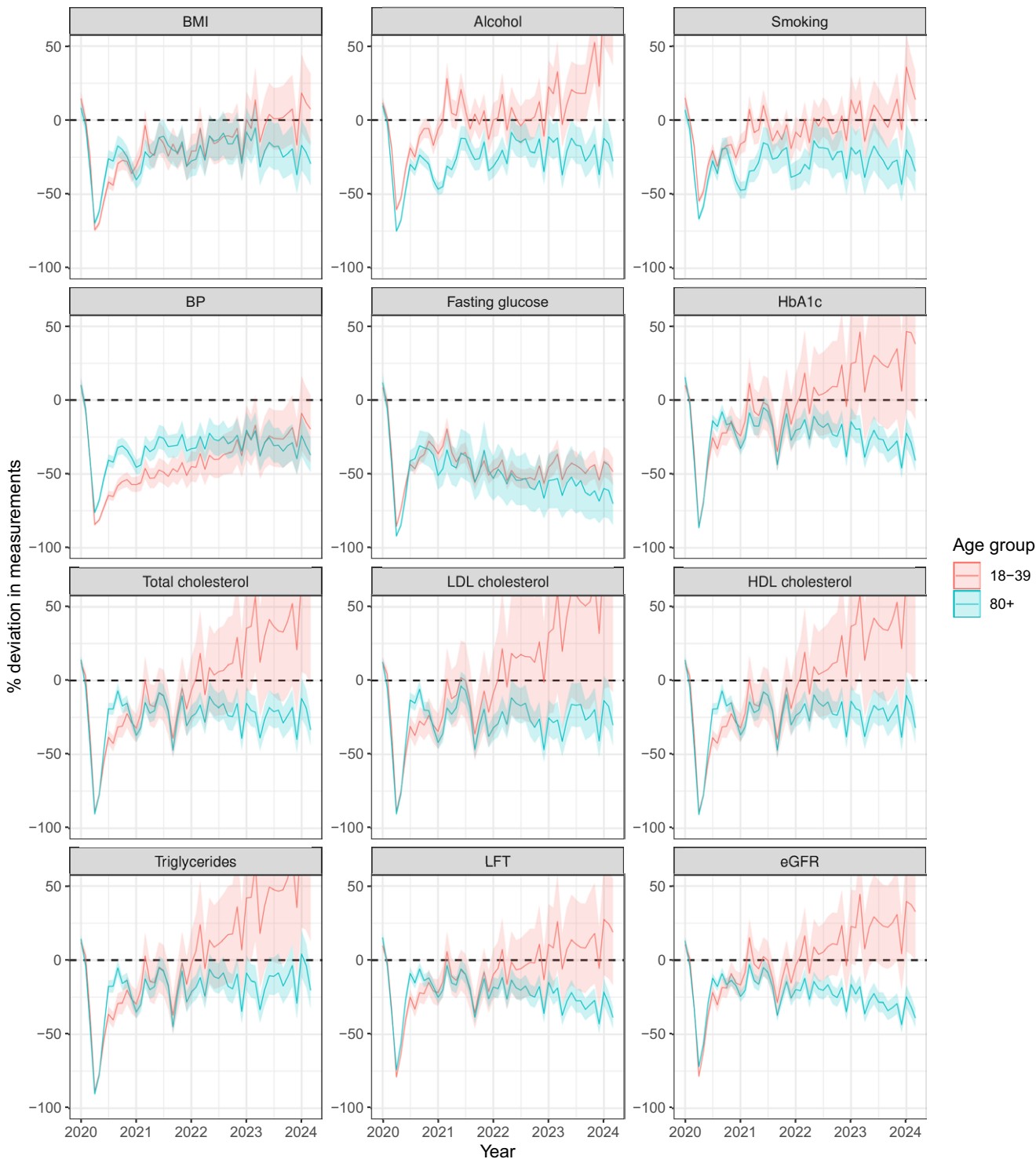

**Fig 3. Estimated % deviation in number of measurements by age groups between November 2018 and March 2024.** The % deviation was estimated as observed—expected/expected. Horizontal dashed line indicates on difference from expected level. Shaded areas are composite of 95% confidence intervals. BMI, body mass index; BP, blood pressure; HbA1c, glycated haemoglobin; LDL, low-density lipoprotein; HDL, high-density lipoprotein; LFT, liver function test; eGFR, estimated glomerular filtration rate.

**Table 2. Estimated deviations in BP measurements overall and by subgroups in March 2023–March 2024.**

| | Number per 1,000 | % |
|---|---|---|
| **Overall** | −18.7 (−36.9, −0.5) | −16.3 (−31.9, −0.6) |
| **Age group** | | |
| 18–39 | −17.8 (−34.2, −1.4) | −23.4 (−43.6, −3.2) |
| 40–59 | −7.2 (−23.5, 9.2) | −4.9 (−23.3, 13.4) |
| 60–79 | −26.5 (−46.9, −6.1) | −16.6 (−28.3, −4.9) |
| 80+ | −57.1 (−87.2, −27.0) | −30.9 (−42.9, −18.8) |
| **Sex** | | |
| Female | −27.6 (−50.6, −4.5) | −19.0 (−33.6, −4.3) |
| Male | −18.8 (−42.1, 4.4) | −12.5 (−29.6, 4.6) |
| **Ethnicity** | | |
| White | −18.2 (−35.5, −0.9) | −22.3 (−39.8, −4.7) |
| Asian | −21.5 (−39.9, −3.0) | −18.7 (−42.8, 5.4) |
| Black | −25.5 (−54.7, 3.8) | −16.0 (−54.6, 22.5) |
| Mixed | −28.2 (−66.7, 10.2) | −16.5 (−49.6, 16.5) |
| Other/unknown | −11.4 (−26.7, 3.9) | −15.1 (−29.2, −1.0) |
| **IMD quintile** | | |
| 1 | −19.8 (−40.5, 0.9) | −16.1 (−33.3, 1.1) |
| 2 | −22.6 (−41.3, −4.0) | −20.4 (−35.6, −5.1) |
| 3 | −19.6 (−36.9, −2.3) | −17.5 (−32.1, −2.8) |
| 4 | −16.2 (−33.1, 0.7) | −14.1 (−29.2, 0.9) |
| 5 | −14.6 (−31.9, 2.7) | −12.5 (−28.5, 3.6) |

Numbers shown were best estimates where parentheses showed 95% CIs combined from 2 estimates of expected levels. IMD, index of multiple deprivation.

than in male (−13%; −19 per 1,000) (Table 2). White people generally had fewer measurements relative to expected compared with other ethnic groups (Tables 2 and C in S2 File). For example, in BP measurements, White people had −22.3% deviation from expected, compared to 15.1% to 18.7% in other ethnic groups.

Deviations in measurements appeared to be J-shaped by the IMD quintile. For example, there was a dose-response reduction in deviation of BP measurements (Fig 4A) from the second most deprived quintile (−20%; −23 per 1,000), third most deprived quintile (−18%; −20 per 1,000), fourth most deprived quintile (−14%; −16 per 1,000), to the least deprived quintile (−13%; −15 per 1,000). People in the most deprived quintile had BP measurement deviation (−16%; −20 per 1,000) were similar to that of those in third quintile. Similarly, but to a lesser extent, relative deviations in HbA1c measurements (Fig 4B) were largest in the 2 most deprived quintile (Q1 most deprived: −2%; Q2: −4%) compared to Q3 (−0.3%), Q4 (+3%), and Q5 (least deprived, +5%).

## Discussion

The present data suggest cardiometabolic risk factor measurements were substantially reduced during when COVID-19–related restrictions were in force. Most of the included risk factor measurements were back to the expected level by 2022, but the recovery of the measurement of BP was prolonged. Importantly there appears to be an inequality in the recovery of risk factor measurements by age, sex, and IMD.

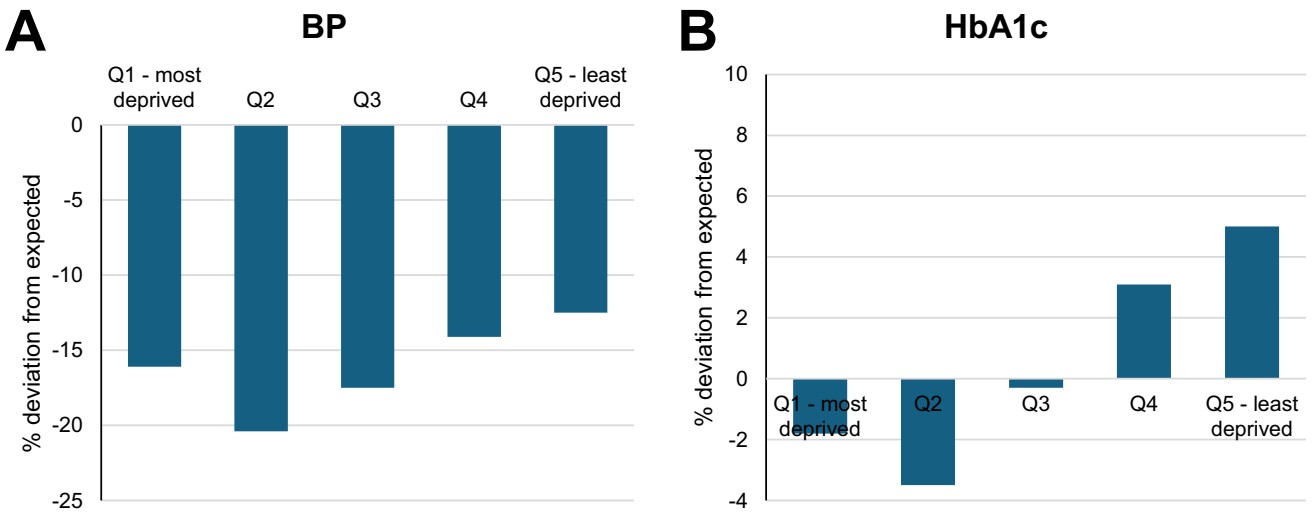

**Fig 4. Estimated % deviation in measurement of BP and HbA1c by IMD quintile in March 2023–March 2024.** BP, blood pressure; HbA1c, glycated haemoglobin; IMD, index of multiple deprivation.

The findings are generally consistent with, but meaningfully extend, previous studies, many focused on the early effects of lockdown. For example, a study quantified the reduction in CVD risk monitoring during the first lockdown [4]. Other studies highlighted the backlog in testing and diagnosis of type 2 diabetes during the early stages of the pandemic [5,6]. An OpenSafely study adopted a similar approach and examined 11 key indicators in general practice [3], some of which, e.g., BP, cholesterol, HbA1c, LFTs, overlaps with the current study. The original paper only covered data up to end of 2021 but the OpenSafely dashboard (https://reports.opensafely.org/reports/opensafely-sro-key-measures-dashboard/) included data comparable to this study. All overlapped risk factors showed similar patterns, adding confidence to the findings of this study. However, this study provided additional information compared the observed number of risk factors with the expected levels based on both the projected trend using retrospective data, and an assumed stable trend. Importantly, we provided additional breakdown by population subgroups to examine the existence of inequality.

Some studies also reported ethnic and social inequality in measurements. A study focused on measurement of HbA1c up to the end of 2021 found strong association between IMD with HbA1c testing, unexplained by diabetes prevalence and proportion of older people [13]. We found a similar trend in HbA1c in our study even though our analysis showed that such inequality still exists by early 2024, and that the association was in a J-shaped pattern. Unlike our study which found slightly larger deficit from expectation in White people, an OpenSafely study concluded ethnic differences in clinical monitoring and remained largely unchanged [14]. There are multiple reasons to explain such difference. Firstly, the OpenSafely study only included ethnicity based on primary care data which was found to be incomplete [7], while the present study used data from linked hospital record to supplement. Secondly, OpenSafely only focused on patients with existing diabetes or CVD, whereas our study included the whole population. It should, however, be noted, like the 2 previous studies, that the present estimation is a comparison with the pre-pandemic trend and therefore could only investigate whether there were any changes from the pattern prior to the pandemic, rather than any existence in inequality per se.

The results of this study should be interpreted with reference to the following limitations. Firstly, due to the time-based cut-off applied to SNOMED-CT codes for the selected risk factors, the initial recorded measurements date from April 2018 to October 2018. Therefore, any estimated long-term trends might not be reliable. Instead, we also projected a stable trend from pre-pandemic to estimate deviations from the pre-pandemic level. Secondly, this study censored people who died during the follow-up and therefore time trend could, in part, reflect a combination of ageing effect (people in need of more risk factor measured as they age) and survival bias (a larger proportion of older people who died during COVID-19 [15]). The large deviation from expected levels in people aged ≥80 could simply reflect that the older people who survived were those with fewer long-term conditions and required fewer risk factor measurements. Thirdly, the GDPPR data only included people who have an active registration with GP on and after 1 November 2019. People who have never registered with a GP or have died between November 2018 and October 2019 have been excluded due to data availability. The GDPPR also does not capture deregistration or migration. Fourthly, the ascertainment of risk factor measurements was based on the GDPPR data, which contains a subset of all SNOMED-CT codes (https://digital.nhs.uk/coronavirus/gpes-data-for-pandemic-planning-and-research/guide-for-analysts-and-users-of-the-data#code-clusters-and-content). The GDPPR includes the most commonly used codes for risk factors related to cardiometabolic factors; hence, the influence of the absence of any codes on our conclusions should be minimal. The codes being used are based on existing National Diabetes Audit or Quality Outcomes Framework code clusters which should be reliable but there might be less used risk factors codes that were missed. Fifthly, the risk factor data were routinely collected during clinical consultations and not for research purposes. It is possible that artifacts may exist within the data due to differences in collection and processing or transfer, and these may vary over time and by source. This study also could not identify the appropriateness of the risk factor measurements and if they are required for disease prevention. Lastly, the trends for primary and secondary prevention might be different, and this was not examined in this study.

This study has important policy and clinical implications. These patterns may in large part explain the reduced number of CVD medications prescribed during the pandemic [2]. Rates of BP measurement have not fully returned to pre-pandemic levels or trends, suggesting that many people with CVD risk factors are potentially being missed. It is particularly worrying that the measurements of BP had a prolonged recovery up until early 2024, given that it is among the highest ranked risk factors for the burden of disease globally [16]. BP is independent risk factors for CVD and can often be managed through combinations of lifestyle and pharmaceutical interventions [17,18]. Indeed, an analysis of over 1.5 million individuals from 8 geographic regions across the world identified systolic blood pressure as contributing the highest population attributable fraction of all CVD risk factors [19]. Importantly, it appears that the reversal in the reduction of risk factors measurements was smaller, or slower, in the younger population under 39 years of age than those aged 40 to 59, among whom prevention could have yielded more substantial benefits over the longer term [20]. Prolonged decrease in risk factor measurements also might indicate a need to compensate the deficit with more measurements of some risk factors now which we could observe in some (e.g., alcohol, triglycerides) but not the others (e.g., BMI, BP, HbA1c). Lack of action to detect these undiagnosed people may lead to a prolonged legacy of potentially preventable cardiometabolic and other diseases linked to these risk factors. Importantly, the findings in this study are corroborated with the latest national health survey in England which showed a marked increase in untreated hypertension (but not in diabetes), rising from 12% in 2019 to 15% in 2021 [21]. The findings could also, in part, explain the sustained excess cardiovascular mortality in the UK [22].

The reasons behind the reductions in BP could not be identified in this study but it could be related to the partial substitution of face-to-face visits by telephone or virtual consultations [23], backlog in primary care [24], and the staffing issues in the NHS [25]. While there have been reports showing telephone consultations could achieve similar effectiveness as face-to-face visits in several surrogate endpoints (e.g., healthcare utilisation, readmission) [23], there has been no evidence, to our knowledge, that evaluated the quality of care (e.g., monitoring of risk factor) and disease endpoints in an unselected population. The preference of telephone and virtual consultations should be cautioned against if these implicate a reduced measurements of important risk factors, especially if such deficiencies are greater in lower socioeconomic areas: such groups are less likely to own home blood pressure monitors or scales, for example. Regardless of the reasons behind, the prolonged period of reduction in measurements represents a large cumulative deficit in risk ascertainment. This likely have hindered preventative efforts as relevant lifestyle advice and medications cannot be administered if people's risk factors are not appropriately identified. Since blood pressure is relevant to many important health outcomes, many risks may be being missed. Rectifying this might require health systems to reintroduce more face-to-face visits wherever possible. Further investigations will be required to understand the reasons behind so that relevant intervention can be designed.

The pandemic could also have accelerated changes in practice related to blood testing. For example, blood glucose tests require fasting and are often conducted in the morning to reduce patient burden. However, with more remote consultations, general practitioners appear to be opting for measuring HbA1c instead of blood glucose. It should be noted that there are debates as to whether diagnoses based on HbA1c and fasting glucose are interchangeable [26,27]; HbA1c appears to be the best measurement for diabetes regarding identifying cardiorenal risk [28], though may misclassify a small proportion of individuals.

In summary, using routinely collected primary care data in England, we have shown substantial reductions in rates of CVD risk factor measurements during COVID-19, and a socioeconomically patterned recovery. Further studies should examine whether the identified inequality by age, sex, and area deprivation could be explained by the clinical needs they require. The clinical consequences of having fewer risk factor measurements for a prolonged period of time should also be studied.

## Supporting information

**S1 STROBE checklist. STROBE Statement—Checklist of items that should be included in reports of *cohort studies*.**
(DOC)

**S1 File. Supporting Figures.** Fig A. Flowchart of individual exclusions. Fig B. Monthly number of measurements per 1,000 individuals by age group. Fig C. Monthly number of measurements per 1,000 individuals by sex. Fig D. Monthly number of measurements per 1,000 individuals by ethnicity. Fig E. Monthly number of measurements per 1,000 individuals by IMD quintile. Fig F. Estimated % deviation in number of measurements by sex. Fig G. Estimated % deviation in number of measurements by ethnicity. Fig H. Estimated % deviation in number of measurements by IMD quintile.
(PDF)

**S2 File. Supporting Tables.** Table A. Results from generalised additive models. Table B. Number (per 1,000) deviation in risk factor measurements by sociodemographic subgroup in 3 periods. Table C. % deviation in risk factor measurements by sociodemographic subgroup in 3 periods.
(XLSX)

## Acknowledgments

This study makes use of de-identified data held in NHS England's Secure Data Environment service for England and made available via the BHF Data Science Centre's CVD-COVID-UK/COVID-IMPACT consortium. This work uses data provided by patients and collected by the NHS as part of their care and support. We would also like to acknowledge all data providers who make health relevant data available for research, and Mr Rouven Priedon for administrative support.

CVD-COVID-UK/COVID-IMPACT Consortium includes Frederick K Ho, Caroline Dale, Mehrdad A Mizani, Thomas Bolton, Ewan R Pearson, Jonathan Valabhji, Chris Tomlinson, Steffen E Petersen, Benjamin Bray, Mark Ashworth, Kazem Rahimi, Mamas Mamas, Julian Halcox, Cathie Sudlow, Reecha Sofat, Naveed Sattar and others.

## Author Contributions

**Conceptualization:** Frederick K. Ho, Caroline Dale, Reecha Sofat, Naveed Sattar.

**Data curation:** Frederick K. Ho, Mehrdad A. Mizani, Thomas Bolton.

**Formal analysis:** Frederick K. Ho.

**Investigation:** Caroline Dale, Mehrdad A. Mizani, Thomas Bolton, Ewan R. Pearson, Jonathan Valabhji, Christian Delles, Paul Welsh, Shinya Nakada, Daniel Mackay, Jill P. Pell, Chris Tomlinson, Steffen E. Petersen, Benjamin Bray, Mark Ashworth, Kazem Rahimi, Mamas Mamas, Julian Halcox, Cathie Sudlow, Reecha Sofat, Naveed Sattar.

**Writing – original draft:** Frederick K. Ho, Naveed Sattar.

**Writing – review & editing:** Caroline Dale, Mehrdad A. Mizani, Thomas Bolton, Ewan R. Pearson, Jonathan Valabhji, Christian Delles, Paul Welsh, Shinya Nakada, Daniel Mackay, Jill P. Pell, Chris Tomlinson, Steffen E. Petersen, Benjamin Bray, Mark Ashworth, Kazem Rahimi, Mamas Mamas, Julian Halcox, Cathie Sudlow, Reecha Sofat.

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
