## [Editor Report · Decision Letter 0]

22 Jan 2024

Dear Dr Ho, 

Thank you for submitting your manuscript entitled "Measurement of routine cardiometabolic disease risk factors in primary care following Covid-19 pandemic" for consideration by PLOS Medicine.

Your manuscript has now been evaluated by the PLOS Medicine editorial staff and I am writing to let you know that we would like to send your submission out for external peer review.

Please re-submit your manuscript within two working days, i.e. by Jan 24 2024 11:59PM.

Feel free to email me at pdodd@plos.ord or the team at plosmedicine@plos.org if you have any queries relating to your submission.

Kind regards,

Pippa

Philippa Dodd, MBBS MRCP PhD

PLOS Medicine

pdodd@plos.org

---

## [Decision Letter · Decision Letter 1]

14 May 2024

Dear Dr. Ho,

Many thanks for submitting your manuscript "Measurement of routine cardiometabolic disease risk factors in primary care following Covid-19 pandemic, PMEDICINE-D-24-00197R1” to PLOS Medicine. The paper has been reviewed by two subject experts and a statistician; their comments are included below and can also be accessed here:

[LINK]

As you will see, the reviewers were positive about the paper but, they raised a number of questions about specific study details and the methodological approach. After discussing the paper with the editorial team and an academic editor with relevant expertise, I’m pleased to invite you to revise the paper in response to the reviewers’ comments. We plan to send the revised paper to some of all of the original reviewers*, and of course we cannot provide any guarantees at this stage regarding publication.

When you upload your revision, please include a point-by-point response that addresses all of the reviewer and editorial points, indicating the changes made in the manuscript and either an excerpt of the revised text or the location (eg: page and line number) where each change can be found. Please submit a clean version of the paper as the main article file and a version with changes marked should as a marked-up manuscript. Please also check the guidelines for revised papers at http://journals.plos.org/plosmedicine/s/revising-your-manuscript for any that apply to your paper.

We ask that you submit your revision by June 4th 2024. However, if this deadline is not feasible, please contact me by email, and we can discuss a suitable alternative.

Please don’t hesitate to contact me directly with any questions (pdodd@plos.org). If you reply directly to this message, please be sure to ‘Reply All’ so your message comes directly to my inbox.

Kind regards,

Pippa

Philippa Dodd MBBS MRCP PhD

PLOS Medicine

plosmedicine.org

pdodd@plos.org

*Please note: If your article is accepted, you may have the opportunity to make the peer review history publicly available. The record will include editor decision letters (with reviews) and your responses to reviewer comments. If eligible, we will contact you to opt in or out.

Editorial comments:

1) We liked your paper and agree with the reviewers that it offers a helpful addition to the existing data regarding the impact of the pandemic and the trajectory of the post-pandemic recovery of health services in the UK. As currently presented it is not without limitations, as identified by the reviewers, which we require you address prior to further consideration.

2) Data Availability – thank you for including a statement which requires some revision. 

As the data are not freely available, please describe briefly the ethical, legal, or contractual restriction that prevents you from sharing it. Please also include an appropriate contact (web or email address) for inquiries. Please note that the point of contact cannot be a study author.

The information detailed on pages 9/10 (Ethical & data access approvals) could be placed in the submission form under the data availability sub-section.

3) Data reporting - please ensure that the study is reported according to the STROBE guideline, and include the completed STROBE checklist as Supporting Information. Please add the following statement, or similar, to the Methods: "This study is reported as per the Strengthening the Reporting of Observational Studies in Epidemiology (STROBE) guideline (S1 Checklist)."

When completing the checklist, please use section and paragraph numbers, rather than page and/or line numbers as these often change in the event of publication.

4) Protocol/statistical analysis plan – did your study have a prospective protocol or analysis plan? Please state this (either way) early in the Methods section.

For all observational studies, in the manuscript text, please indicate: (1) the specific hypotheses you intended to test, (2) the analytical methods by which you planned to test them, (3) the analyses you actually performed, and (4) when reported analyses differ from those that were planned, transparent explanations for differences that affect the reliability of the study's results. If a reported analysis was performed based on an interesting but unanticipated pattern in the data, please be clear that the analysis was data-driven.

5) Author Summary - at this stage, we ask that you include a short, non-technical Author Summary of your research to make findings accessible to a wide audience that includes both scientists and non-scientists. The authors summary should consist of 2-3 succinct bullet points under each of the following headings:

• Why Was This Study Done? Authors should reflect on what was known about the topic before the research was published and why the research was needed.

• What Did the Researchers Do and Find? Authors should briefly describe the study design that was used and the study’s major findings. Do include the headline numbers from the study, such as the sample size and key findings. 

• What Do These Findings Mean? Authors should reflect on the new knowledge generated by the research and the implications for practice, research, policy, or public health. Authors should also consider how the interpretation of the study’s findings may be affected by the study limitations. In the final bullet point of ‘What Do These Findings Mean?’, please describe the main limitations of the study in non-technical language.

The Author Summary should immediately follow the Abstract in your revised manuscript. This text is subject to editorial change and should be distinct from the scientific abstract. Please see our author guidelines for more information: https://journals.plos.org/plosmedicine/s/revising-your-manuscript#loc-author-summary

6) Introduction – this is currently a little slender. Please ensure that you address past research and explain the need for and potential importance of your study. Indicate whether your study is novel and how you determined that. If there has been a systematic review of the evidence related to your study (or you have conducted one), please refer to and reference that review and indicate whether it supports the need for your study.

7) Discussion - please ensure that you present and organize the Discussion as follows: a short, clear summary of the article's findings; what the study adds to existing research and where and why the results may differ from previous research; strengths and limitations of the study; implications and next steps for research, clinical practice, and/or public policy; one-paragraph conclusion.

Comments from the reviewers:

Reviewer #1: Summary: This large, population-level study used primary care data from England to describe trends in the measurement of 14 cardiometabolic risk factors from April 2019 to April 2023. It showed approximately 5.52 million fewer risk factor measurements per month than expected between March 2020 and February 2022 during pandemic restrictions, especially among those aged under 60 years. After February 2022, reductions in BMI, BP and Hba1c measurements persisted.

Originality/ importance: This study shows the power of large electronic health record datasets for analyses across a suite of cardiometabolic risk factor measures to assess pandemic impact on care. While many other studies report reductions in GP consultations for routine clinical monitoring during the pandemic, few have extended to post-pandemic time periods. There has been one large OpenSAFELY study reporting similar patterns across 11 indicators of GP clinical activity up to December 2021, accompanied by a dashboard of measures for monitoring GP clinical activity that is currently updated to December 2023. Nevertheless, the present study is a more focussed deep dive into cardiometabolic risk factor recording by age-group, sex and deprivation, with important implications for public health.

Comments

1. Check the wording of the title: an article seems to be missing

2. Under 'Evidence before this study', a key omission is the OpenSAFELY study of trends in GP clinical activity in England by Fisher et al(1). This should be referenced.

3. The methods section should clarify who is in the study population: was it adults only? At which timepoint(s) were they assessed for eligibility? It was also unclear why those who had died by April 2023 were excluded completely rather than being censored at death.

4. The study design should be clearly stated in the methods.

5. The section describing the cardiometabolic risk factors was brief. It would be helpful to include further rationale for the choice of these risk factors and explanation about how and when they were measured.

6. Currently, the link to the pre-specified analysis plan on GitHub does not work so it is not possible to compare the pre-specified analysis plan with results presented.

7. In the results section, information is needed on characteristics of the study population. The total number of individuals is given as 631 million - is this a typo?

8. In the supplementary tables, an explanatory footnote would aid interpretation of the betas.

7. The first sentence of the discussion is grammatically incorrect, as is the sentence in paragraph two beginning 'Regardless of the reasons.' The language of the discussion should be reviewed throughout.

8. There was some inconsistency in interpretation of the findings related to socioeconomic status (SES): while text in the results suggested little difference in measurements by SES, except for cigarette smoking, the interpretation cautioned against virtual consultations to avoid widening inequalities among groups without access to home blood pressure monitors, especially if such deficiencies are greater in lower socioeconomic areas. However, this is not what the study found.

9. In the mechanisms section, it was suggested that obesity can lead to cardiovascular disease through Hba1c. Glycated haemoglobin is a marker of blood glucose levels over the last 2-3 months and can indicate poor diabetic control, rather than a mechanism leading towards cardiovascular disease in itself. The language around this needs to be clarified.

10. The paragraph on consistency with existing literature should mention the OpenSAFELY study and dashboard findings on routine GP clinical activity including cardiovascular risk factor measurement.

11. Further explanation is needed in the limitations about why results obtained from extrapolation of the pre-pandemic trend might be 'somewhat biased' and 'less relevant to the post-pandemic population.'

References

(1). Fisher et al. Eleven key measures for monitoring routine general practice clinical activity during COVID-19: a retrospective cohort study using 48 million adults' primary care records in England through OpenSAFELY. Elife 2023; 21:e84673. 

Dashboard: https://reports.opensafely.org/reports/opensafely-sro-key-measures-dashboard/

Reviewer #2: This is an interesting study on the change of number of measurements of routine cardiometabolic disease risk factors in primary care following Covid-19 pandemic. However, there are a couple of major issues needing attention.

1) Counterfactual estimates. We can see from Figure 1 that the observed numbers of measurements were mostly recovered to pre-pandemic levels or following the previous trends. The key findings are in Figure 2 following the adjustment/substraction from the counterfactual estimates. Now the robustness and reliability of the counterfactual estimates become absolutely crucial in concluding that there is reduction in number of measurement. How these counterfactual estimates are exactly derived? from only these 4 years data? Normally data from longer time such as 10 to 20 years are needed to establish trends for prediction and also with all the potential predictors and risk factors. Do we have these? In other words, how reliable and robust are these counterfactual estimates? 

2) In the stats analysis section, it says "Measurements of people who had died by April 2023 were excluded to eliminate competing risk". However, this is not the way to handle competing risk. The results could be biased because of this as it only applies to people alive which excluded around 200k people died in the pandemic in the UK.

Reviewer #3: Thank you for inviting me to review this manuscript which estimates the number of missed tests for cardiometabolic risk factors following the onset of the COVID-19 pandemic. The main finding is a significant reduction in the number of tests recorded at the onset of the pandemic, with incomplete recovery as of April 2023. These findings are a useful extension to existing literature which can help inform the continued post-pandemic recovery. However, there are a few areas of concern and some small clarifications required before recommending the manuscript for publication, highlighted below.

Major issues

* The authors provide a link to an associated GitHub repo but it is not public. This should be made public for review. 

* The codelists used to identify the various risk factor measurements are unclear. A link is provided, but this leads to the GDPPR guide for analysts. Can the authors provide a more specific link? Or highlight where they can be found in the GitHub repo?

* As the authors highlight, there was an increasing trend observed across the measurements prior to the pandemic. In some measurements, the extent of this is quite surprising. For example, there is a doubling of blood pressure measurements in the first year of the study period. This is not consistent with references 3 and 20, which show more stable rates of measurements prior to the pandemic for HbA1c, cholesterol and blood pressure. If this is not a true reflection of actual activity, it is likely the subsequent estimated number of missed measurements is overestimated. Can the authors provide any explanation for this increase? 

 - The authors describe the removal of measurements for people who have died by April 2023. Is it possible that the positive trend, at least in part, reflects an increasing denominator population 

 as new patients can enter the study population throughout the study period? Could you instead include measurements for those that die before the end of the study period and calculate the 

 rate with person-months as the denominator and see if the trend persists?

Minor issues

* Title

 - Include the setting

 - The title suggests this study is restricted to the post-pandemic period. 

 - A suggested revision:

 "Impact of the COVID-19 pandemic on the routine measurement of cardiometabolic disease risk factors in primary care in England"

* Abstract

 - Findings. It would be useful to state the number of patients impacted by missed measurements as well as the total number of missed measurements.

* Intro

 - "Modelling predicted that this decline could result in an excess of over 13,000 CVD events in the Great Britain". It's not obvious this is referring to citation 2. Recommend citing again here.

* Methods

 - It's evident from the age groups used that this study is restricted to adults but you could more clearly state this in the text.

* Results

 - Including the total number of rows in GDPPR is of limited use for this study. The use of "primary care record" here could be confusing as this is more commonly used to refer to the complete 

 set of recordings for an individual. 

 - You could be clearer on what the 712 million represents. This is the total number of risk factor measurements in the study population between April 2019 and April 2023. 

 - You state that these measurements are across 631 million individuals. This is presumably missing a decimal point. If so, this is at odds with the 57 million in the Databases section, presumably 

 due to patients meeting the age criteria during the study period, which may be worth describing.

 - The order of magnitude used for the number of measurements isn't very easy to work with. It would be easier if these were presented in either thousands or millions.

 - Table 1. Related to the above, the figure title states "number per '00,000" which suggests this is a rate.

 - Figure 1. 

 - The data represented is from Apr 2019-23, but the lines appear to start before 2019. I suggest aligning the x ticks with the start of the year rather than the middle to avoid confusion.

 - This could also be helped by including the date period in the figure title.

 - Indicate what the shaded area represents in the legend or figure title.

 - Results paragraph 4 - include the absolute number as well as the percentage reduction, as in the previous paragraph.

[LINK]

1. Please upload any figures associated with your paper as individual TIF or EPS files with 300dpi resolution at resubmission; please read our figure guidelines for more information on our requirements: http://journals.plos.org/plosmedicine/s/figures. While revising your submission, please upload your figure files to the PACE digital diagnostic tool, https://pacev2.apexcovantage.com/. PACE helps ensure that figures meet PLOS requirements. To use PACE, you must first register as a user. Then, login and navigate to the UPLOAD tab, where you will find detailed instructions on how to use the tool. If you encounter any issues or have any questions when using PACE, please email us at PLOSMedicine@plos.org.

To submit your revised manuscript please use the following link:

---

## [Decision Letter · Decision Letter 2]

23 Sep 2024

Dear Dr. Ho,

Thank you very much for re-submitting your manuscript "Impact of the COVID-19 pandemic on the decline and recovery of routine measurement of cardiometabolic disease risk factors in primary care in England" (PMEDICINE-D-24-00197R2) for review by PLOS Medicine.

I have discussed the paper with my colleagues and the academic editor and it was also seen again by 2 reviewers. I am pleased to say that provided the remaining editorial and production issues are dealt with we are planning to accept the paper for publication in the journal.

[LINK]

We look forward to receiving the revised manuscript by Sep 30 2024 11:59PM.   

Kind regards,

Pippa

Philippa Dodd, MBBS MRCP PhD

Senior Editor 

PLOS Medicine

plosmedicine.org

GENERAL

Thank you very much for your detailed and considered responses to previous comments. Please see below for further comments which we require that you address in full prior to publication.

Please note that some of the requests may not apply to your study design and some may have already been incorporated into the manuscript but please review the complete list and include any additional items as necessary.

COMMENTS FROM THE ACADEMIC EDITOR

I am happy to support your decision to accept the manuscript. I have two general comments that you may wish to consider when providing feedback to the authors.

First, I am not entirely convinced by the authors’ argument that pressure measurements did not return to baseline. Upon reviewing Figure 2, it appears that the measurements have largely returned to baseline, despite some periodic variability that may explain the fluctuations in confidence intervals. Additionally, I find the claim regarding fasting glucose levels not returning to baseline unpersuasive. The pre-COVID trend shows a sharp decrease, likely due to a shift towards hemoglobin A1c testing that you can kind of see in the pre-pandemic data. I believe the paper works well as a descriptive analysis, and I would encourage the authors to focus more on the broader trends and disparities related to COVID-19, rather than emphasizing the deficits in return to baseline, which the data does not seem to robustly support.

Second, while the authors appropriately identify key limitations in the discussion, I believe they could be prioritized differently. In my view, the most significant limitations are (1) the limited amount of pre-COVID data available for predictive modeling, and (2) potential data artifacts related to mortality, which may alter the denominator and introduce bias.

COMMENTS FROM THE EDITORS:

TITLE

Please revise your title according to PLOS Medicine's style. Your title must be nondeclarative and not a question. It should begin with main concept if possible. "Effect of" should be used only if causality can be inferred, i.e., for an RCT. Please place the study design ("A randomized controlled trial," "A retrospective study," "A modelling study," etc.) in the subtitle (ie, after a colon). 

We suggest, “Routine measurement of cardiometabolic disease risk factors in primary care in England during the COVID-19 pandemic: A population-based cohort study“ or similar.

DATA AVAILABILITY STATEMENT

Please also include the following details in the data availability section of the manuscript submission form when you resubmit:

* Details regarding your analysis code including the URL (and any accession codes) i.e., https://github.com/BHFDSC/CCU008_01

* A URL for e BHF Data Science Centre’s CVD-COVID-UK/COVID-IMPACT consortium and a contact email address for a Mr Rouven Priedon.

* These details from pages 9 & 10 of the PDF: ‘The data used in this study are available in NHS England’s Secure Data Environment (SDE) service for England, but as restrictions apply they are not publicly available (https://digital.nhs.uk/services/secure-data-environment-service). The CVD-COVIDUK/COVID-IMPACT programme led by the BHF Data Science Centre (https://bhfdatasciencecentre.org/) received approval to access data in NHS England’s SDE service for England from the Independent Group Advising on the Release of Data (IGARD) (https://digital.nhs.uk/about-nhs-digital/corporate-information-and-documents/independentgroup-advising-on-the-release-of-data) via an application made in the Data Access Request Service (DARS) Online system (ref. DARS-NIC-381078-Y9C5K) (https://digital.nhs.uk/services/data-access-request-service-dars/dars-products-andservices). The CVD-COVID-UK/COVID-IMPACT Approvals & Oversight Board (https://bhfdatasciencecentre.org/areas/cvd-COVID-uk-COVID-impact/) subsequently granted approval to this project to access the data within NHS England’s SDE service for England. The de-identified data used in this study were made available to accredited researchers only as per the data sharing agreement and the ethical approval. Those wishing to gain access to the data should contact bhfdsc@hdruk.ac.uk in the first instance.

ABSTRACT

Please structure your abstract using the PLOS Medicine headings (Background, Methods and Findings, Conclusions).

Please combine the Methods and Findings sections into one section, “Methods and findings”.

Abstract Methods and Findings:

Please ensure that all numbers presented in the abstract are present and identical to numbers presented in the main manuscript text.

Please include the study design, population and setting, number of participants, years during which the study took place, length of follow up, and main outcome measures.

Please quantify the main results (with 95% CIs and p values).

Please include the important dependent variables that are adjusted for in the analyses.

Please include the actual amounts and/or absolute risk(s) of relevant outcomes (including NNT or NNH where appropriate), not just relative risks or correlation coefficients. (example for absolute risks: PMID: 28399126). 

Please include a summary of adverse events if these were assessed in the study.

In the last sentence of the Abstract Methods and Findings section, please describe the main limitation(s) of the study's methodology.

AUTHOR SUMMARY

Thank you for including an author summary which reads very nicely. Please see below for some minor suggestions:

* Why Was This Study Done?

Bullet point #2 suggest: ‘However, the extent to which recovery has occurred throughout 2024, remains unknown as do outcomes according to age, sex, ethnicity or deprivation.’ Or similar

* What Did the Researchers Do and Find?

Bullet point #1: suggest, ‘…which covers…’

Bullet point #2: suggest, ‘…risk factor…’

* What Do These Findings Mean?

Please include a final bullet point of ‘What Do These Findings Mean?’, please describe the main limitations of the study in non-technical language.

METHODS and RESULTS

Please note the comments above from the academic editor.

Please move the ethics statement from page 9 and include in the methods section.

TABLES and FIGURES

Throughout the main manuscript, please name figure labels using Arabic numerals, and abbreviate the word “Figure” to “Fig” (e.g., Fig 1, Fig 2, Fig 3, etc.

Please Cite figures with the format: Fig 1, Fig 2, Fig 3, etc.

Please ensure that all abbreviations are defined in the caption or an appropriate footnote, including those used to report statistical information.

Please ensure to include the meaning of any dots/lines/bars.

Please consider avoiding the use of red and green in order to make figures more accessible to those with colour blindness.

To help facilitate transparent data reporting, where adjusted analyses are presented please also present the unadjusted analyses for comparison. In a caption or footnote please detail all factors adjusted for.

DISCUSSION

Please amend the sub-heading to read ‘Discussion’ (as opposed to Discussions).

Please remove all information on page 9/10 except the acknowledgements and include only in the relevant sections of the manuscript submission form and methods as detailed above (DATA AVAILABILITY STATEMENT).

REFERENCES

For in-text reference callouts, please place citations in square as opposed to semi-circular brackets.

SUPPORTING INFORMATION

In the published article, supporting information files are accessed only through a hyperlink attached to the captions. For this reason, you must list captions at the end of your manuscript file. You may include a caption within the supporting information file itself, as long as that caption is also provided in the manuscript file. Do not submit a separate caption file.

When supplementary files are contained with a single file: 

Please label the file as ‘S1 Supporting Information’.

Please apply alphabetical labelling to each table and figure contained within the S1 file. For example, ‘Fig A’ to ‘Fig Z’ and ‘Table A’ to ‘Table Z’.

Plain text does not need to be labelled and can just be given a title as necessary. For example, ‘Statistical Analysis Plan’.

Please cite tables/figures as ‘Fig A in S1 Supporting Information’ and/or ‘Table A in S1 Supporting Information’, for example.

Please cite plain text as, ‘Statistical Analysis Plan in S1 Supporting Information’, for example.

Alternatively when supplementary files are uploaded as separate files:

Please label tables as ‘S1 Table’ (so on)

Please label figures as ‘S1 Fig’ (and so on)

Any additional documents (protocols/analysis plans etc.) can be labelled as ‘S1 Protocol’, for example.

Please cite items as exactly as labelled.

STROBE Checklist – thank you for including the checklist. Please amend to refer to section and paragraph numbers as opposed to page (or line) numbers as the latter often change at the time of publication.

SOCIAL MEDIA

To help us extend the reach of your research, please detail any X (formerly Twitter) handles you wish to be included when we tweet this paper (including your own, your coauthors’, your institution, funder, or lab) in the manuscript submission form when you re-submit the manuscript.

Comments from Reviewers:

Reviewer #1: The manuscript has been substantially strengthened in this revised version. I am satisfied that my comments have been adequately addressed. 

Reviewer #2: Thanks authors for their effort to improve the manuscript. I am satisfied with the response and revision. No further questions from me.

[LINK]

---

## [Editor Report · Decision Letter 3]

4 Oct 2024

Dear Dr Ho, 

On behalf of my colleagues and the Academic Editor, Dr David Flood, I am pleased to inform you that we have agreed to publish your manuscript "Routine measurement of cardiometabolic disease risk factors in primary care in England before, during and after the COVID-19 pandemic: A population-based cohort study" (PMEDICINE-D-24-00197R3) in PLOS Medicine.

Please change 'vary' to 'varies' in the author summary when you make your formatting changes (as detailed below).

PRESS

Thank you again for submitting to PLOS Medicine. It has been a pleasure handling your manuscript and we look forward to publishing your paper. 

Kind regards,

Pippa

Philippa C. Dodd, MBBS MRCP PhD 

Senior Editor 

PLOS Medicine

pdodd@plos.org